# Cryo-EM structure of 5-HT$_{3A}$ receptor in its resting conformation

Sandip Basak [1], Yvonne Gicheru[1], Amrita Samanta[1], Sudheer Kumar Molugu [2], Wei Huang[2],
Maria la de Fuente[2], Taylor Hughes[2], Derek J. Taylor[2], Marvin T. Nieman[2], Vera Moiseenkova-Bell [1,2] &
Sudha Chakrapani[1,3]

Serotonin receptors (5-HT$_{3A}$R) directly regulate gut movement, and drugs that inhibit 5-HT$_{3A}$R function are used to control emetic reflexes associated with gastrointestinal pathologies and cancer therapies. The 5-HT$_{3A}$R function involves a finely tuned orchestration of three domain movements that include the ligand-binding domain, the pore domain, and the intracellular domain. Here, we present the structure from the full-length 5-HT$_{3A}$R channel in the apo-state determined by single-particle cryo-electron microscopy at a nominal resolution of 4.3 Å. In this conformation, the ligand-binding domain adopts a conformation reminiscent of the unliganded state with the pore domain captured in a closed conformation. In comparison to the 5-HT$_{3A}$R crystal structure, the full-length channel in the apo-conformation adopts a more expanded conformation of all the three domains with a characteristic twist that is implicated in gating.

[1] Department of Physiology and Biophysics Case Western Reserve University Cleveland OH 44106-4970 USA. [2] Department of Pharmacology, Case Western Reserve University, Cleveland, OH 44106-4970, USA. [3] Department of Neuroscience, School of Medicine, Case Western Reserve University, Cleveland, OH 44106-4970, USA. Correspondence and requests for materials should be addressed to S.C. (email: Sudha.chakrapani@case.edu)

The ion channel class of serotonin receptors (5-HT$_3$R) are cation selective and belong to the super family of pentameric ligand-gated ion channels (pLGICs)[1,2]. The 5-HT$_3A$R pentameric complex is ~250 KDa in molecular weight and is made up of five homologous subunits A (5-HT$_3A$) or a heterologous combination of subunit A with other subunits (B–E) arranged around a pseudo five-fold symmetric axis[3]. These channels, located in the dorsal vagal complex of the brainstem and in the gastrointestinal (GI) tract, form a circuit that controls gut motility, visceral perception, secretion, and the emesis reflex[4,5]. 5-HT$_3A$R are implicated in a number of psychiatric and GI disorder conditions including anxiety, depression, bipolar disorder, and irritable bowel syndrome[6,7]. Currently, serotonin receptor (5-HT$_3A$R) antagonists are in clinical use to alleviate nausea and vomiting caused by chemotherapy and radiotherapy, and for the management of post-infection diarrhea and irritable bowel syndrome[8,9]. However, in several cases, adverse side effects have led to restrictions in use of these drugs[10]. A better understanding of the structural correlates of 5-HT$_3A$R function will therefore facilitate ongoing drug design efforts for safer therapeutics.

At the functional level, the 5-HT$_3A$R gating cycle involves transitions between at least three distinct conformational states, namely: the resting state, a non-conductive conformation with low agonist-affinity; the open state, a conductive conformation with higher agonist-affinity; and the desensitized state, a non-conductive conformation, with the highest agonist-affinity among the three states. In the absence of the agonist (serotonin), the channel resides predominantly in the resting or closed conformation, while in the presence of the agonist, the channel transiently opens and eventually transitions to the desensitized conformation. Several therapeutically interesting compounds act as orthosteric or allosteric ligands and modulate 5-HT$_3A$R channel activity by shifting the equilibrium between these pre-formed functional states. At the structural level, 5-HT$_3A$R is a dynamic allosteric protein where binding of the neurotransmitter serotonin in the N-terminal extracellular domain (ECD) elicits a conformational change leading to pore opening within the transmembrane domain (TMD). In addition to these two domains, the channel has a large intracellular domain (ICD) formed by the region between the transmembrane M3 and M4 helices. Although high resolution structures of several prokaryotic and eukaryotic pLGICs are now available[11–19], the key questions regarding the conformational coupling between the ECD, TMD, and the ICD still remain unclear. The ICD is implicated to play a role in receptor trafficking and clustering at the synapse plasma membrane, gating, post-translational modification, and intracellular regulation of channel function[20]. Additionally, in 5-HT$_3A$R, the relatively large ICD also modulates single-channel channel conductance, rectification, and desensitization kinetics[21,22]. In the pLGIC structures solved so far, the ICD is intrinsically missing, or has been genetically removed or considerably digested by trypsin[13–17]. In the trypsin-digested 5-HT$_3A$R crystal structure, the ICD was partially resolved as two split α-helices; a short horizontal MX helix extending from the post-M3 loop and the MA helix extending from the cytoplasmic side toward M4[16]. Moreover, the 5-HT$_3A$R was crystallized in the presence of single-chain antibodies (referred to as nanobodies) that bound to the channel in the vicinity of the serotonin binding site. Functional analysis showed that in the nanobody-bound form, the channel is not activated by serotonin, and hence the channel conformation likely corresponds to that of an inhibited, non-conducting channel[16]. Although the inhibited state (at least in the case of competitive antagonism) could be expected to resemble the resting (closed) conformation, experimental evidences from voltage-clamp fluorometry, X-ray crystallography, and theoretical predictions

in pLGIC homologs suggest that the inhibitors elicit conformational changes of their own even though the channel is electrically silent[23–27]. Interestingly, the antagonist-induced structural changes at some positions are similar while others distinct to those evoked by the agonist suggesting that the antagonist stabilizes the receptor in a different non-conductive conformation[23,25]. Thus, to determine the basic tenets of the resting conformation, we sought to determine the structure of the full-length 5-HT$_3A$R channel in the apo-conformation by single-particle cryo-electron microscopy (cryo-EM). The cryo-EM structure of apo-5-HT$_3A$R at 4.3 Å resolution is similar to the 5-HT$_3A$R crystal structure but is captured in a distinct state with differences in the ECD, TMD, and the ICD conformations.

## Results

**Function and the cryo-EM structure of 5-HT$_3A$R.** The full-length mouse 5-HT$_3A$R was cloned into an oocyte expression vector for functional measurements. In response to application of serotonin, robust concentration-dependent macroscopic inward currents were elicited that desensitize in the continued presence of the activating ligand (Supplementary Fig. 1A). A plot of peak response as a function of serotonin concentration yielded an EC$_{50}$ of $2.7 \pm 0.09\,\mu M$ which is in agreement with the previous studies[28] (Supplementary Fig. 1A). For structural studies, the full-length 5-HT$_3A$R gene was codon-optimized, cloned into pFast-Bac1 plasmid, and expressed in *Spodoptera frugiperda* (Sf9) cells. Milligram quantities of 5-HT$_3A$R samples were purified to homogeneity. The purified protein was deglycosylated with PNGase treatment and gel-filtration fractions corresponding to pentamers (Supplementary Fig. 1B) were used for single-particle cryo-EM analysis (Supplementary Fig. 1C). Three dimensional reconstructions of the detergent-solubilized, full-length 5-HT$_3A$R channel led to a density map with a nominal resolution of 4.3 Å with imposed C$_5$ symmetry (Supplementary Figs. 1D, 2A, and 2B). The final reconstruction was built from 108,727 particles (Supplementary Fig. 3) and had a local resolution in most regions between 3.5 and 4.5 Å (Supplementary Fig. 1D). The map contained density for the entire ECD, TMD, and a large region of the ICD, and was used for model building and refinement (Supplementary Fig. 4). The final refined apo-5-HT$_3A$R model contains residues Asp6-Gln338 and L397-Ser461 (Table 1).

When viewed from the plane of the bilayer, perpendicular to the pore axis, the apo-5-HT$_3A$R pentamer is 165 Å in length and has a diameter of 80 Å (Fig. 1a, b). The ECD reveals protruding non-protein densities corresponding to N-glycans. The mouse 5-HT$_3A$R has three N-glycosylation sites at positions Asn82, Asn148, and Asn164, and strong densities for glycans were observed at each of the asparagine sidechains. Site-directed mutagenesis shows that prevention of glycosylation at sites Asn82, Asn148, and Asn164 abolishes the expression of functional receptors on the membranes[29]. Densities of the glycans were also resolved in several previous pLGIC structures including 5-HT$_3A$R, GABA-β3, nAChR, GlyR, and GluCl[13,16,17,30]. The overall topology of apo-5-HT$_3A$R is as previously observed in other members of the pLGIC family with each subunit comprised of a twisted β-sandwich ECD and a four-helical TMD. In addition, in apo-5-HT$_3A$R, the density for the ICD is only partially resolved with post-M3 loop, MX, and MA helices visible.

We then compared the apo-5-HT$_3A$R conformation with that of 5-HT$_3A$R crystal structure[16]. An alignment of the two structures reveals an expansion of all the three domains in apo-5-HT$_3A$R resulting from a radial outward twist of the ECD outer β-sheet and an iris-like movement of the TM helices and the ICD (Fig. 2a). The ECD undergoes a clockwise twist with respect to the

**Table 1 Data collection and processing**

| Data collection/processing | 5-HT$_{3A}$R Apo |
|---|---|
| Microscope | Titan Krios |
| Voltage (kV) | 300 |
| Defocus range (μM) | −0.75 to −2.5 |
| Exposure time (s) | 12 |
| Dose rate (e⁻/pixel/s) | 4 |
| Pixel size (Å) | 0.532 |
| Number of images | 3550 |
| Number of frames/image | 40 |
| Initial particle number | 327,329 |
| Final particle number | 108,727 |
| Symmetry | C5 |
| Resolution (unmasked, Å) | 4.7 |
| Resolution (masked, Å) | 4.3 |
| Map sharpening B-factor (Å²) | −238 |
| *Refinement* | |
| *Number of atoms* | |
| Protein | 163,49 |
| Ligands | 548 |
| *B factors (Å²)* | |
| Protein | 122 |
| Glycan (NAG) | 110.9 |
| Glycan (BMA) | 140.7 |
| Chloride | 97.6 |
| Sodium | 92.22 |
| Water | 73.6 |
| Lipid | 154 |
| *R.M.S deviations* | |
| Bond length (Å) | 0.00 |
| Bond angle (°) | 1.05 |
| C-beta deviations | 0 |
| *Ramachandran plot* | |
| Favored (%) | 92.91 |
| Allowed (%) | 7.09 |
| Disallowed (%) | 0 |
| Molprobity score | 1.82 (84th percentile) |
| Molprobity clashscore | 4.19 (96th percentile) |

crystal structure, which changes the relative positions of the interfacial loops at the ECD–ECD and ECD–TMD interfaces (Fig. 2b). Particularly, the movement in the outer β-strands is more pronounced than the inner β-strands. In the TMD, the M3 and M4 helices show an outward displacement with respect to an axis through the intrasubunit helical bundle composed of M1–M4 helices, while there is very little change in the position of the M2 helix (Fig. 2b). Changes in M3 and M4 conformations relative to M2 lead to a widening of the intrasubunit cavity lined by M1–M4 helices (indicated by dotted lines, Fig. 2b). These structural differences are also reflected as an increase in total surface area (protein-only region) of the apo-5-HT$_{3A}$R pentamer (102,850 Å²) in comparison to the crystal structure (85,000 Å²). The increase in total surface area is accompanied by an increase in solvent accessible areas for apo-5-HT$_{3A}$R pentamer (73,210 Å²) in comparison to the crystal structure (54,530 Å²). The analysis of inter-subunit interfaces reveals that the buried areas between adjacent principal and complementary subunits are slightly reduced in apo-5-HT$_{3A}$R (2858 Å²) compared to the crystal structure (3045 Å²). A comparison of different pLGIC structures captured in various conformational states seems to suggest that activation involves a concerted counter-clockwise twist of the ECD around the pore axis[15,19,30]. It was previously noted that the nanobody-bound 5-HT$_{3A}$R is twisted counter-clockwise with respect to the pLGIC putative closed conformations such that it lies further along the pathway that defines the global motion going from closed to open structures[16]. In apo-5-HT$_{3A}$R, the ECD twist is clockwise with respect to the crystal structure, and in the direction toward the closed conformation.

**The ion permeation pathway and the TMD.** Viewing from the extracellular end (Fig. 3a, b), the ion permeation pathway in apo-5-HT$_{3A}$R is wide open with a radius greater than 3.5 Å throughout the ECD followed by a narrower pathway in the TMD and the ICD with multiple constriction sites (Supplementary Fig. 5). A slight constriction is seen at the center of the ECD lined by Lys108 in the β4–β5 loop, although the extent of constriction is less than that seen in the crystal structure (Fig. 3b). A negative charge in the vicinity of Lys108 (at position Asp105) has been shown to regulate single-channel conductance in pLGICs[31]. The Asp105 position is conserved among most cation-selective pLGICs, whereas this position is a lysine in anion-selective members of the family[32].

The pore of apo-5-HT$_{3A}$R is lined by M2 which appear as straight helices lying parallel to the pore axis (Fig. 3c). At the mouth of the extracellular end, the pore is lined by Asp20′, followed by three rings of hydrophobic residues at Ile17′, Val13′, and Leu9′. Below Leu9′, the pore is polar lined by Thr6′, Ser2′, and negatively charged at the intracellular end with Glu-1′. The Asp-4′ is positioned further away from the pore axis. The radius of the pore at Asp20′ is 3.8 Å and decreases to 2.1 Å at the level of Leu9′ position. Strong density (7–6 σ) and weaker density (5–4 σ) are present within the permeation pathway in the TMD and the ICD (Fig. 3c). The weaker densities are arranged as two sets of five-membered rings: one as a loose arrangement, in-plane with Asp20′ sidechain with a mean distance of 3.94 Å and the other as a tight pentagon ring beneath it with a mean distance of 2.4 Å. These values are close to those estimated for water polygons in high-resolution protein crystals[33]. Based on these distances, the pore radii at this location (>hydrated Na⁺ ion radius), and the B-factors for the modeled ions and waters (Table 1), we suggest that these densities likely correspond to ions and water. A similar arrangement of ions and water pentagons was also observed in the intracellular end of M2 in the GLIC open structure[34]. Further down, the pore at Leu9′ is constricted to ~2.3 Å which is too small to allow a hydrated Na⁺[35] to pass through and in addition the hydrophobicity of this region prevents ion dehydration. There are two additional constrictions below the Leu9′ position, at Thr6′ and Ser2′ with a radius of 2.7 Å and 2.1 Å, respectively. Although the pore radii at these locations are smaller than a hydrated Na⁺, the hydroxyl sidechain at these positions may allow coordination of partially dehydrated ions. The intracellular end is lined by two rings of negative charges at positions Glu-1′ and Asp-4′ which are involved in governing charge selectivity and constitute the selectivity filter[36]. This suggests that the channel in this conformation is non-conducting and the Leu9′ position acts as a gate impeding ion permeation. This is not surprising considering that the channel was imaged in the absence of any activating ligands. The finding that the activation gate is located at the center of M2 is also in agreement with SCAM studies in 5-HT$_{3A}$R which showed that extracellularly positioned Cys in M2, particularly the 17′ position, were accessible even in the closed state. In contrast, residues below the 13′ are accessible in the open but not in the closed state[37]. In the strychnine-bound GlyR structure, the channel is constricted primarily at the extracellular end with constrictions to below 3 Å radii at 20′, 13′, and 9′ positions[15]. The "closed" GLIC conformation is also much narrower in the extracellular end with positions 16′ and 17′ constricted to 2 Å[19]. On the other hand, the 5-HT$_{3A}$R structures are similar to the apo-GluCl structure with a relatively wider pore at the external end[30] (Supplementary Fig. 6).

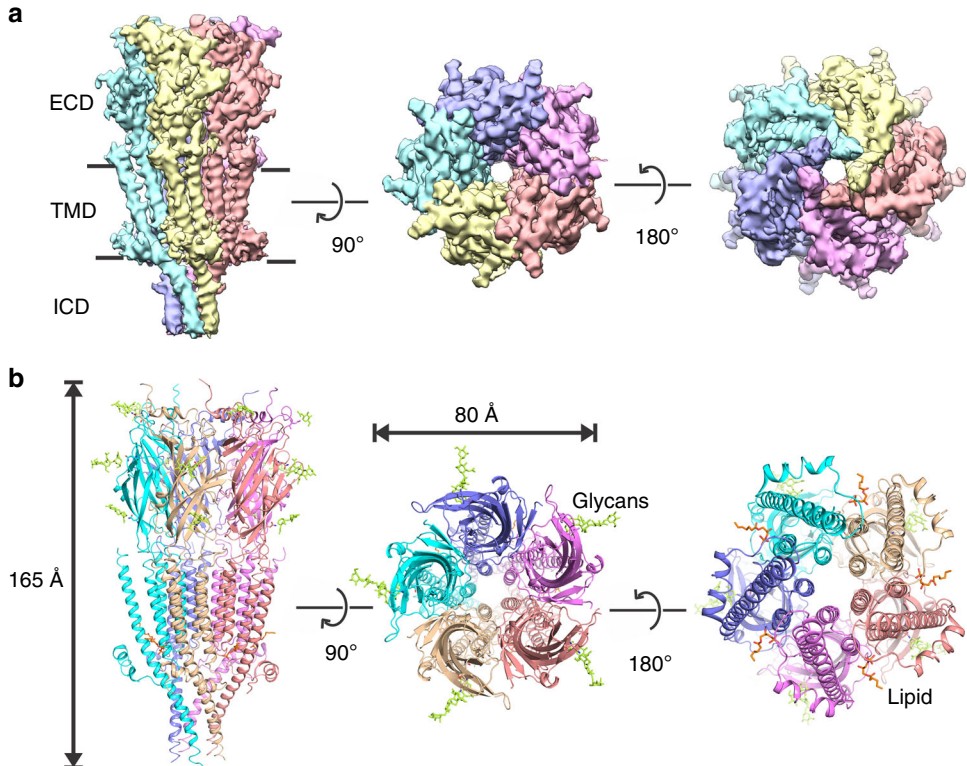

**Fig. 1** Cryo-EM structure of apo-5-HT$_{3A}$R. **a** The 3D reconstruction map from the full-length 5-HT$_{3A}$R at 4.3 Å resolution. The views, going from left to right, are parallel to the membrane (side view), from the extracellular side (top view), and from the intracellular side (bottom view). Individual subunits are depicted in different colors, and the three domains are labeled. The solid lines denote putative membrane limits. **b** Cartoon representations of the 5-HT$_{3A}$R structural model based on the EM reconstruction. The views correspond to the orientations shown in **a**. For each subunit, three sets of glycans (green) and one lipid (brown) molecule are shown as stick representation

At the extracellular end, M2 extends into the M2–M3 linker, which in apo-5-HT$_{3A}$R is bent and tilted upward (above the level of the cys loop), such that it is in close proximity with the pre-M1 region as well as the β8–β9 loop of the neighboring subunit (Supplementary Fig. 7A). Residue Ala277 in the M2–M3 linker makes a H-bond contact with Arg219 in the pre-M1 region, while Thr276 interacts with the backbone carbonyl of Gly185 in the β8–β9 loop. The M2–M3 linker was only partially resolved in the 5-HT$_{3A}$R crystal structure[16]. The M2–M3 linker is seen in multiple orientations in different pLGIC structures with this region adopting a distinct extended conformation in the apo-5-HT$_{3A}$R (Supplementary Fig. 7B). The M2–M3 linker in the agonist-bound GlyR structure is retracted from the inter-subunit interface in comparison to the antagonist-bound structure[15]. The M2–M3 linker and the pre-M1 region are implicated to be a part of the signal transduction machinery that communicates ligand-binding events in the ECD to the channel gate in the M2 bundle[38]. Consequently, mutations in these regions are shown to impact channel function[39].

The M4 segment has a well-defined density and the tip of M4 is α-helical extending above the putative membrane limits (Supplementary Fig. 8). The tip of M4 is in close proximity to the β6–β7 (Cys) loop with Trp459 in M4 oriented toward the conserved Phe144 in the Cys loop in a potential stacking interaction. The tip of M4 was not well-resolved in the 5-HT$_{3A}$R crystal structure and Trp459 was built only in two subunits[16]. Of the two, one of the Trp sidechains was in a similar stacking orientation. In nAChR, the interaction between the post-M4 and cys loop is predicted to modulate gating[40].

**Conformation of the ECD**. The neurotransmitter binding site has been extensively studied and a wealth of structural data is available from the acetylcholine binding protein (AChBP) and from the 5-HT$_{3A}$-AChBP chimera bound to a battery of pLGIC agonists and antagonists[24,41]. The serotonin binding site is located at the subunit interface lined by residues from Loops A, B, and C on the (+) subunit and Loops D, E, and F from the (−) subunits[42]. There are clear densities for aromatic residues (Fig. 4a) from each of these loops lining the binding pocket, particularly Trp156-Loop B, Phe199 and Tyr207-Loop C, Trp63-Loop D, Tyr126-Loop E. Densities for Trp168, Arg169, and D177 in Loop F are also shown and are implicated in ligand binding. In comparison to the crystal structure, the β-sheets forming the ECD undergo a clockwise twist (viewed from the extracellular end) (Fig. 4b) leading to a widening of the ligand-binding pocket with rearrangement of the binding site residues (Fig. 4c). In addition, Loop C, which is at the periphery of the neurotransmitter binding pocket, in the apo-5-HT$_{3A}$R structure is placed outward in an open or "extended" conformation. The Loop C conformation has been implicated to correlate both with the occupancy of the ligand at the neurotransmitter binding site and the functional state of the channel[24]. Particularly in AChBP, in comparison to the apo-state, in the antagonist-bound state, Loop C adopts an "open" conformation, while in the agonist-bound state it takes a "closed" conformation. A partly "closed" conformation of Loop C in the crystal structure is likely from its interaction with the nanobody bound in the vicinity. The orientation of Trp156 in Loop B, lining the back wall of the neurotransmitter binding pocket, in apo-5-HT$_{3A}$R is flipped relative to its position in the

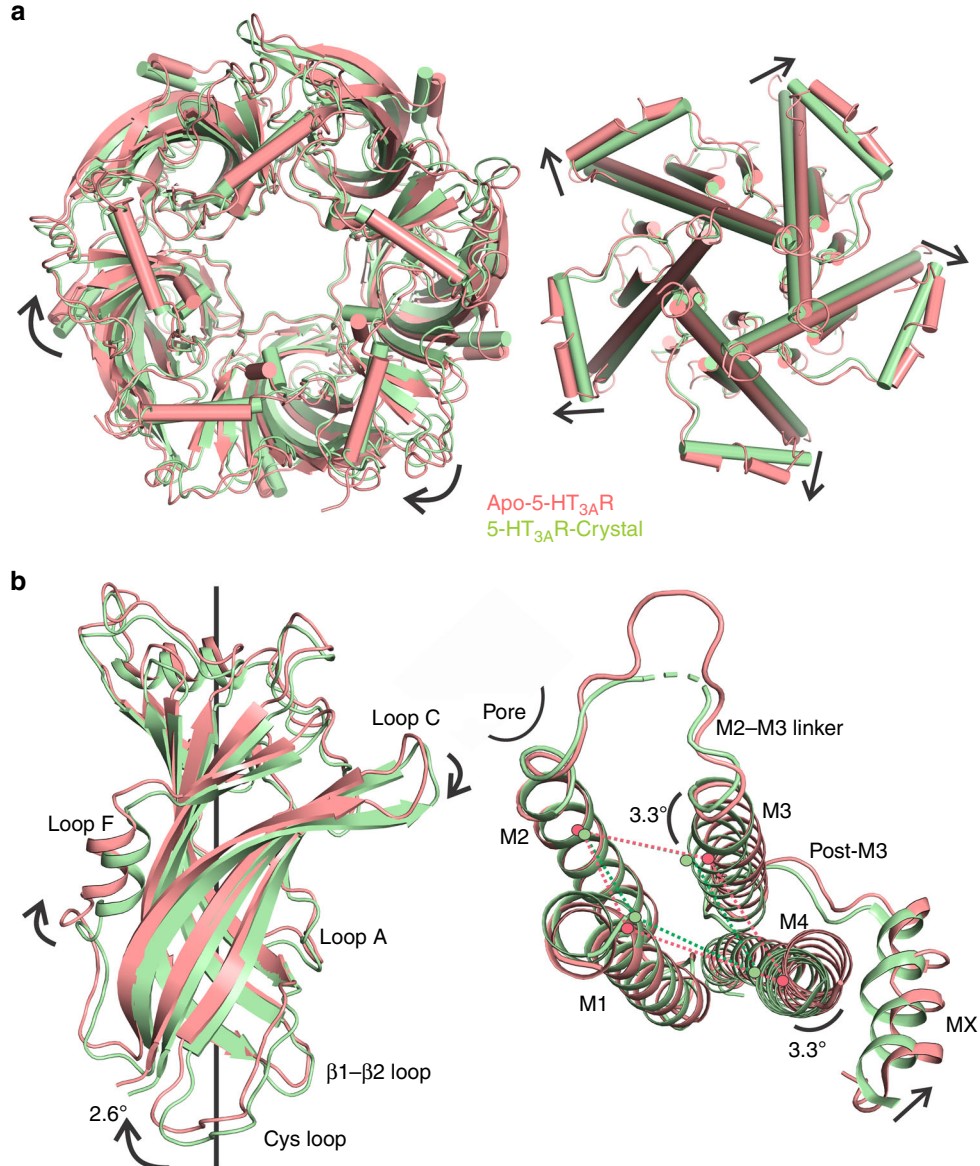

**Fig. 2** Alignment of the apo-5-HT$_{3A}$R with the crystal structure of nanobody-bound 5-HT$_{3A}$R. **a** A view of the ECDs from the extracellular end when aligned with respect to the TMDs (left). The view of the TMDs from the intracellular end when aligned with respect to the ECDs (right). The apo-5-HT$_{3A}$R structure and the 5-HT$_{3A}$R crystal structure are shown in salmon red and pale green, respectively. The arrows show the putative direction of displacements between the two structures. **b** A comparison of the ECD of the apo-structure with the crystal structure when aligned with respect to the TMD of the (−) subunit (left). A comparison of the TMD between the two structures when aligned with the ECD of the (−) subunit. The relative tilt of the axis parallel to the TM helices between the two structures are indicated. The dotted lines highlight the differences in the intrasubunit cavity volume. The spheres indicate the position of residues M1 (Leu227), M2 (Leu266), M3 (Met291), and M4 (Trp456). The arrows show the putative direction of displacements between the two structures (right). The alignment in **b** highlights the relative changes in the two structures, both with respect to the neighboring subunit, as well as with respect to the other domain

5-HT$_{3A}$R crystal structure. This position is highly conserved as a Trp in 5-HT$_{3A}$R, α-subunits of nAChRs, and AChBP, while it is a Tyr in the anionic members of pLGIC family. The indole ring of Trp is involved in a cation-π interaction with the quaternary ammonium ligands binding to the neurotransmitter binding pocket[43]. An alignment with AChBP and nAChR shows that the Trp156 orientation in apo-5-HT$_{3A}$R is similar to the conformation seen in these structures (Supplementary Fig. 9). Further studies are necessary to understand the potential role of Trp156 re-orientation in channel 5-HT$_{3A}$R inhibition.

**Conformation of the ICD.** The ICD sequence is the least conserved region among the pLGIC family, and in human isoforms

the length is always greater than 70 residues and varies up to 150 residues. In apo-5-HT$_{3A}$R, the post-M3 region forms a loop that extends away from the pore axis and terminates in an α-helix referred to as MX helix. MX is a short helix which forms a belt around the MA helical bundle, lying parallel to the plane of the membrane, and is seen to bend downward toward the cytoplasmic region (Fig. 5a). Although we see densities for the entire MX and MA helices and for a short region of the unstructured portion from MX, the density for a stretch of 59 residues is missing until beginning of the MA helix, presumably due to the unstructured and flexible nature of this loop. Not surprisingly, this is also the region that is chipped away during trypsin digestion[16]. The apo-5-HT$_{3A}$R structure suggests the presence of

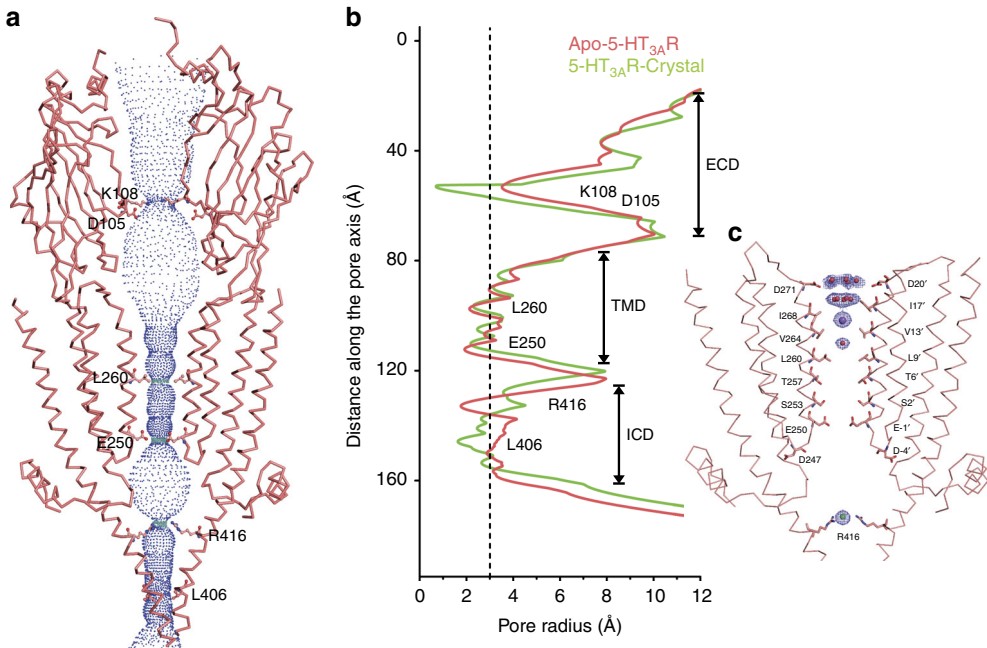

**Fig. 3** Profile of ion permeation pathway. **a** The pore profile generated by the HOLE program[60] depicts an ion permeation pathway of ~165 Å encompassing the ECD, TMD, and the ICD. Only two subunits are shown for clarity. Sidechains of residues that line the constricted areas are shown as sticks. **b** A comparison of the pore radii along the pore axis for the 5-HT$_{3A}$R cryo-EM structure (salmon red) with that of the crystal structure (pale green). The dashed line indicates an approximate radius of a hydrated Na$^+$ ion[35]. The pore is constricted below 3 Å radius at three sites: L260 and E250 along M2 and R416 in the ICD. **c** Non-protein densities along the pore axis were modeled as water (red), Na$^+$ (magenta), and Cl$^-$ (green). The map around the ions is shown as a mesh representation calculated at various σ values (outer water ring: 4 σ; inner water ring:5 σ; Na$^+$ ion: 6 σ; Cl$^-$ ion: 7 σ)

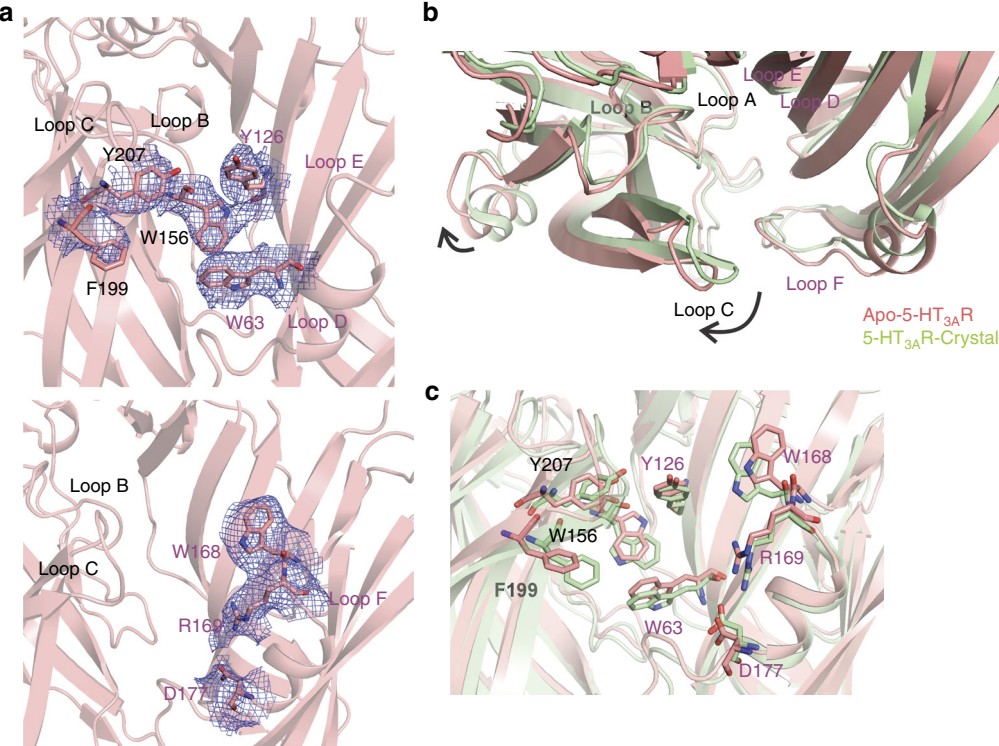

**Fig. 4** The neurotransmitter binding site. **a** The map around the aromatic residues at the subunit interface that constitutes the neurotransmitter binding site (top). The map for the residues in Loop F that are involved in ligand binding (bottom). **b** Alignment of 5-HT$_{3A}$R apo (salmon red) and crystal (pale green) structures reveals a twist and an expansion at the region lined Loop C, Loop B, and Loop F. The arrows indicate the direction of movement. **c** A comparison of the orientations of the residues that are involved in neurotransmitter binding

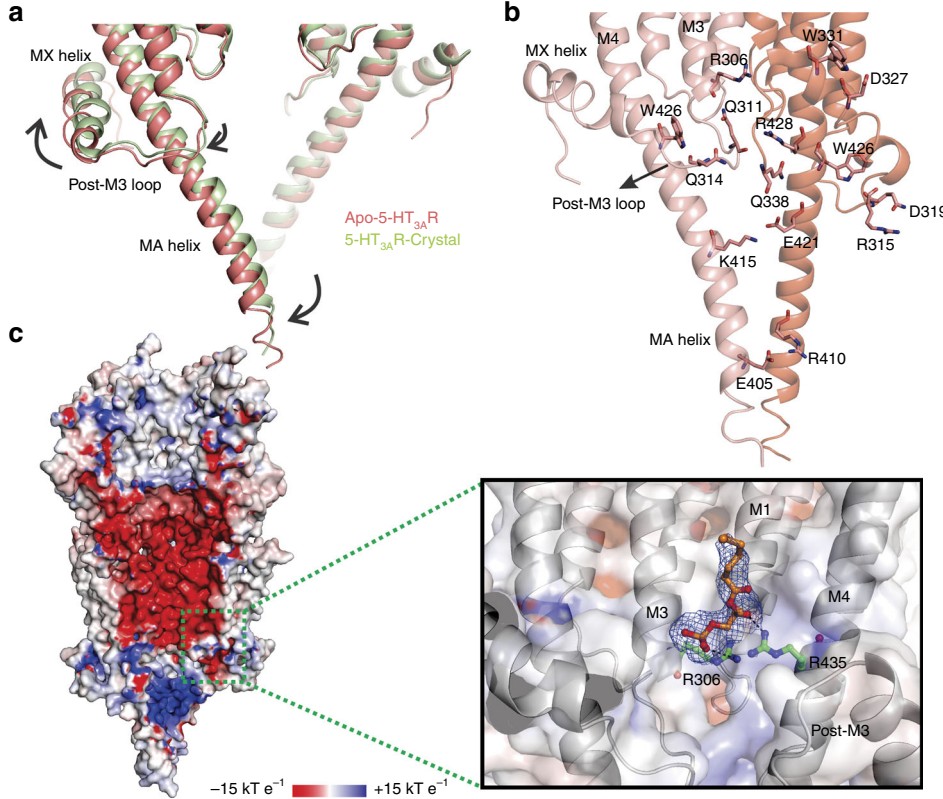

**Fig. 5** The Intracellular domain. **a** The ICD is comprised of the post-M3 loop, the MX, helix, a stretch of unstructured region, followed by the MA helix. The density from the MX helix is bent downward to the intracellular end of the MA helix, but the unstructured region is not resolved. Superposition of the 5-HT$_{3A}$R apo and crystal structures reveals differences in the conformation of ICD in the two structures. The (−) subunit of the two structures are aligned. The expansion of the ICD resulting from an outward displacement of MA and MX helices are indicated by arrows. **b** The residues within the ICD involved in several potential intra and inter-subunit interactions. **c** The solvent-accessible electrostatic potential map generated using the APBS tool. The inset shows a zoomed-in view of the lipid-binding pocket within the dotted green box. The lipid (partially built phospholipid) and the interacting residues (R306 and R435) are shown in stick. The map around the lipid is shown as blue mesh

multiple intrasubunit interactions between MX, unstructured loop, and MA. The post-M3 loop is held in its position through potential polar interactions with M3 (Arg306) and M4 helix (Trp426) within the same subunit (Fig. 5b). Although in comparison to the 5-HT$_{3A}$R crystal structure the post-M3 loop is positioned away from the pore axis (with minimal interaction with the neighboring subunit) (Fig. 5a), it still partially blocks the inter-subunit cavity or lateral portals which have been implicated to serve as conduits for ion permeation[12].

The ICD reveals a notable constriction to 2.1 Å at the level of Arg416 with a strong density (6 $\sigma$) at the center of the pore axis, which we modeled as a Cl$^-$ (Fig. 3c and Supplementary Fig. 5). Interestingly, previous studies have suggested that intracellular phosphates interact with MA arginine residues near the cytoplasmic mouth of the channel facilitating pore block and inhibition of ion conduction[44]. Besides the constriction at Arg416, the MA helix offers a continuous conduit of radius over 3 Å. Serotonin-mediated activation, therefore, likely evokes a conformational change within the ICD to allow permeation. 5-HT$_{3A}$R has an unusually low conductance which is well below the resolution of single-channel recordings. The Arg416 residue is a part of a cluster of arginine residues in the MA helix that are shown to be a critical determinant of 5-HT$_{3A}$R conductance and mutations at these positions lead to significant increases in single-channel conductance[21]. In contrast to the tightly packed arrangement of MA helices in the crystal structure with an obstruction at the hydrophobic patch at the C-terminal end of the helix lined by Leu402, Leu406, Ile409, and Leu413, the apo-5-

HT$_{3A}$R structure reveals a more loose packing of the MA helices with the hydrophobic residues positioned farther away from the pore axis. This finding is in agreement with mutagenesis studies that suggest that this lower region has a lesser impact on channel conductance in comparison to the charged cluster further up[45].

The electrostatic potential map along the ion permeation pathway shows that in contrast to the ECD and the TMD, where the surface charge is electronegative (lined by Asp52, Glu53 (β1-β2 loop), and Glu186 (β8–β9 loop) in the ECD and Glu-2′ and Asp-4′ (M2) in the TMD), the ICD has a cluster of positively charged residues (highlighted by the blue region in Fig. 5c). This is a unique feature of 5-HT$_{3A}$R among the cation-selective pLGICs. At the edge of the TMD–ICD interface, there is a density present in the cavity lined by M3 of one subunit and M1 and M4 helices of the neighboring subunit. This density was partially modeled as a phospholipid based on putative polar contacts with neighboring Arg sidechains (Arg306 in M3 and Arg435 in M4). Interestingly, a recently solved structure of the GABA (β3–α5) chimeric channel reveals a neurosteroid binding site that aligns with the lipid site suggesting that this lipid pocket may be involved in allosteric modulation of pLGICs[46].

## Discussion

The cryo-EM structure of apo-5-HT$_{3A}$R determined in the solution state, in the absence of inhibitors, Fab fragments, truncation, or constraints from crystal lattice, bears the hallmarks of the resting conformation. In comparison to the 5-HT$_{3A}$R crystal structure, the full-length channel adopts an expanded

conformation with a clockwise twist in the β-strands of the ECD and the Loop C in an extended position. In addition, the locations of constriction that occlude ion permeation in the resting conformation can be inferred. The pore-lining region reveals non-protein densities, the arrangement of which suggests that they correspond to ions and water molecules. Further, the structure suggests that the ICD is involved in several inter-subunit interactions that may underscore its role in signal transduction across the channel. If the crystal structure and the apo-5-HT$_3$AR structure were to truly represent physiologically inhibited and resting conformations, respectively, the conformational differences in the ECD, TMD, and ICD provide insights into mechanisms underlying channel modulation. The apo-5-HT$_3$AR structure serves as a starting conformation to determine ligand-induced conformational changes leading to open and desensitized states.

## Methods

**5-HT$_3$AR cloning and electrophysiological measurements in oocytes**. The gene encoding 5-HT$_3$AR (purchased from GenScript USA Inc.) was inserted into a *Xenopus laevis* oocyte expression vector (pTLN) and confirmed by DNA sequencing. This construct was first linearized with *Mlu1* restriction enzyme overnight at 37 °C and then used for mRNA synthesis using the mMessage mMachine kit (Ambion). The mRNA was purified with RNAeasy kit (Qiagen) and stored at −20 °C. For measurement of 5-HT$_3$AR macroscopic currents, *Xenopus laevis* oocytes (stages V–VI) were injected with 3–10 ng of mRNA. As a control, oocytes were injected with the corresponding volume of water to verify endogenous currents were not present. The oocytes used in this study were kindly provided by Dr. Walter F. Boron. Female *X. laevis* were purchased from Nasco. All animal experimental procedures were approved by Institutional Animal Care and Use Committee (IACUC) of Case Western Reserve University. Both sets of injected oocytes were maintained at 18 °C in OR3 media (GIBCO BRL Leibovitz media containing glutamate and supplemented with 500 units each of penicillin and streptomycin, pH 7.5. The osmolarity was adjusted to 197 mOsm). After 2–5 days of injection, two electrode voltage-clamp (TEVC) experiments were performed at room temperature on a Warner Instruments Oocyte clamp OC-725. Oocytes were clamped at −60 mV holding potential, and macroscopic current traces were recorded in response to the application of serotonin hydrochloride (at indicated concentrations). The solution-exchange was performed using a syringe-pump perfusion system operating at a flow-rate of 6 ml/min. The current was sampled and digitized at 500 Hz with a Digidata 1332A. The traces were analyzed by Clampfit 10.2 (Molecular Devices). The electrophysiological solutions had the following buffer composition: 96 mM NaCl, 2 mM KCl, 1.8 mM CaCl$_2$, 1 mM MgCl$_2$, and 5 mM HEPES (osmolarity adjusted to 195 mOsm, pH 7.4). All chemical reagents used in these experiments were purchased from Sigma-Aldrich.

**Cloning and transfection**. Codon-optimized mouse 5-HT$_3$AR gene (NCBI Reference Sequence: NM_001099644.1) was purchased from GenScript USA Inc. Subsequently, it was subcloned into pFastBac1 vector containing four Strep-tags (WSHPQFEK) at the N-terminus, followed by a linker sequence (GGGSGGGSGGGS) and a TEV-cleavage sequence (ENLYFQG). The construct also includes a C-terminal 1D4-tag[47]. *S. frugiperda* (Sf9) cells (purchased from Expression System) were cultured in ESF921 medium (Expression System) and incubated at 28 °C without CO$_2$ exchange. The culture media did not contain antibiotics. Sub-confluent cells were then transfected with recombinant 5-HT$_3$AR bacmid DNA using Cellfectin II transfection reagent (Invitrogen) using manufacturer recommended instructions. At 72 h post-transfection, the cell-culture supernatant was collected and centrifuged at 1000×*g* for 5 min to remove cell debris and harvest progeny 1 (P1) recombinant baculovirus. The P2 virus was obtained through a consecutive round of Sf9 cells infection with the P1 virus stock. The supernatant containing P2 virus was then used to infect Sf9 cells, thus generating P3 virus. The level of protein expression was checked by western blot using both P2 and P3 viruses. The P3 virus was used for recombinant protein production and analysis.

**Expression and purification of recombinant protein**. The recombinant 5-HT$_3$AR protein production was carried out by infection of approximately 2.5 × 10$^6$/ml Sf9 cells with P3 virus. The cells were harvested at 72 h post-infection and centrifuged at 8000×*g* for 20 min at 4 °C to separate the supernatant from the cell pellet. The cells were then re-suspended in a buffer containing 20 mM Tris-HCl and 36.5 mM sucrose at pH 7.5 and supplemented with 1% protease inhibitor cocktail (Sigma-Aldrich). The cells were disrupted by sonication on ice and the non-lysed cells were removed by centrifugation (3000×*g* for 15 min). The membrane fraction was separated by ultracentrifugation (167,000×*g* for 1 h) and solubilized with 1% C$_{12}$E$_9$ (Anatrace) in a buffer containing 500 mM NaCl, 50 mM Tris pH 7.4, 10% glycerol, and 0.5% protease inhibitor by rotating for 2 h at 4 °C. Non-solubilized material

was removed by ultracentrifugation (167,000×*g* and 15 min). The supernatant was collected and bound with 1D4 beads equilibrated with 150 mM NaCl, 20 mM HEPES pH 8.0, and 0.01% C$_{12}$E$_9$ for 2 h at 4 °C. The beads were then washed with 100 column volumes of 150 mM NaCl, 20 mM Hepes pH 8.0, and 0.01% C$_{12}$E$_9$ (Buffer A). The protein was then eluted with Buffer A supplemented with 3 mg/ml 1D4 peptide (NH$_2$-TETSQVAPA-CO$_2$H). Eluted protein was then concentrated and deglycosylated with PNGase F (NEB) by incubating 5 units of the enzyme per 1 µg of the protein for 2 h at 37 °C under gentle agitation. Deglycosylated protein was then applied to a Superose 6 column (GE healthcare) equilibrated with Buffer A. The peak fractions around 13.9 ml were pooled and concentrated to 2–3 mg/ml using 50 kDa MWCO Millipore filters (Amicon) and used subsequently for cryo-EM studies.

**Sample preparation and cryo-EM data acquisition**. The 5-HT$_3$AR protein (~2.5 mg/ml) was filtered and incubated with 3 mM Fluorinated Fos-choline 8 (Anatrace) to improve particle distribution[48]. The sample was double blotted (3.5 µl per blot) onto Cu 300 mesh Quantifoil 1.2/1.3 grids (Quantifoil Micro Tools), and immediately after the second blot, the grid was plunge frozen using a Vitrobot (FEI). The grids were imaged on a Titan Krios microscope (FEI), operating at 300 kV, and equipped with a K2-Summit direct detector camera (Gatan). 40-frame movies were collected at 130,000× magnification (set on microscope) in super-resolution mode with a physical pixel size of 0.532 Å/pixel. The dose rate was 4 electrons/pixel/second, with a total exposure time of 12 s. The defocus values ranged from −0.75 µm to −2.5 µm (input range setting for data collection) as per the automated imaging software[49].

**Image processing**. Movies were motion-corrected to compensate for the beam-induced motion using MotionCor2[50] with a B-factor of 150 pixels$^2$. Super-resolution counting images were 2 × 2 binned in Fourier space with a pixel size of 1.064 Å. All subsequent data processing was performed using RELION 2.03[51]. The defocus values of the motion-corrected micrographs were estimated using Gctf software[52]. Approximately, 2000 particles were manually picked from the 3550 micrographs and sorted into two-dimensional (2D) classes. The best of these classes were then used as templates for auto-picking. A loose auto-picking threshold was selected to ensure no good particles were missed. This resulted in ~327,000 auto-picked particles that were subjected to multiple rounds of 2D classification to remove suboptimal particles. An initial three-dimensional (3D) model was generated from 5-HT$_3$AR crystal structure (PDB code: 4PIR) and low-pass filtered to 60 Å using EMAN2[53]. The best ~117,000 particles were then subjected to 3D auto-refinement, followed by 3D classification into three classes. The best 3D class, containing ~108,000 particles, was subjected to a final round of 3D auto-refinement and post-processing to yield a 5-HT$_3$AR map at an overall resolution of 4.3 Å (calculated based on the gold-standard Fourier shell coefficient (FSC) = 0.143 criterion). In the RELION post-processing step, a soft mask was applied to the two half-maps before the FSC was calculated. The post-processing step also included B-factor estimation and map sharpening. ResMap software[54] was used for estimation of local resolutions.

**5-HT$_3$AR model building**. The 5-HT$_3$AR crystal structure (PDB-ID: 4PIR)[16] was used as an initial model and aligned to the 5-HT$_3$AR cryo-EM map calculated with RELION 2.03. The cryo-EM map was converted to an .mtz format using mapmask and sfall tools in CCP4i software[55]. The mtz map was then used for model building in COOT[56]. In comparison to the 5-HT$_3$AR crystal structure, for each monomer, 15 additional residues were built in our model and sidechains were built for seven additional residues. After initial model building, the model was refined against the EM-derived maps using the phenix.real_space_refinement tool from the PHENIX software package[57], employing rigid body, local grid, NCS, and gradient minimization. This model was then subjected to additional rounds of manual model-fitting and refinement which resulted in a final model-to-map cross-correlation coefficient of 0.774. Stereochemical properties of the model were evaluated by Molprobity[58].

Protein surface area and interfaces were analyzed by using PDBePISA server (http://www.ebi.ac.uk/pdbe/pisa/). To compare apo-5-HT$_3$AR and the 5-HT$_3$AR crystal structure, all ligands, ions, water molecules, and nanobodies (in the crystal structure) were removed from the PDB files. Additional residues in the apo-5-HT$_3$AR structure were also removed before analysis so that surface area comparisons were made between identical construct lengths. Electrostatic surface potential calculations were carried out using the APBS tools plug-in PyMOL[59]. The pore profile was calculated using the HOLE program[60].

**Data availability**. The coordinates of the 5-HT$_3$AR structure and the cryo-EM map have been deposited under PDB ID 6BE1 and EMD-7088 with the wwPDB and EMDB. Data supporting the findings of this manuscript are available from the corresponding author upon reasonable request.

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

## Acknowledgements

We acknowledge the imaging time and the use of instruments at the National Cryo-Electron Microscopy Facility at the NCI. We are grateful to the Cleveland Center for Membrane and Structural Biology for providing the access to the Cryo-EM instrumentation. We thank Denice Major for assistance with hybridoma and cell culture at the

Department of Ophthalmology and Visual Sciences (supported by the National Institutes of Health Core Grant P30EY11373). We thank Dr. Walter F. Boron for kindly providing us with *Xenopus* oocytes and for unrestricted access of the oocyte rig. We are very grateful to the members of the Chakrapani lab for critical reading and comments on the manuscript. This research was, in part, supported by the National Cancer Institute's National Cryo-EM Facility at the Frederick National Laboratory for Cancer Research. This work was supported by the National Institutes of Health grant (1R01GM108921), the Cryo-EM supplement (3R01GM108921-03S1) to S.C., the AHA postdoctoral Fellowship to S.B. (17POST33671152), and the National Institutes of Health grant (1R01GM103899) to V.M.-B.

## Author contribution

S.B. and S.C. conceived the project and designed the experimental procedures. S.B. purified the protein and carried out all aspects of cryo-EM sample preparation, screening, data analysis, model building, and refinement. Y.G. performed the two-electrode voltage-clamp recordings. A.S. and V.Y.M.-B. trained S.B. in freezing samples and provided guidance to optimize grid preparation. S.M. trained S.B. and S.C. in the use of Tecnai TF20 microscope for screening grids. M.F. and M.N. trained S.B. in insect cell expression. W.H., T.H., D.J.T., and V.Y.M.-B. provided guidance at various stages of cryo-EM analysis. S.C. supervised the execution of the experiments, data analysis, and interpretation. S.B. and S.C. drafted the manuscript. All authors reviewed the final manuscript.

## Additional information

**Competing interests:** The authors declare no competing financial interests.

