## [Peer Review File · Nature Communications]

Editorial Note: This manuscript has been previously reviewed at another journal that is not operating a transparent peer review scheme. This document only contains reviewer comments and rebuttal letters for versions considered at Nature Communications. Mentions of prior referee reports have been redacted.

Reviewers' comments:

Reviewer #1 (Remarks to the Author):

In 2014 Hassaine et al. described the X-ray crystal structure of 5-HT_{3A} receptor in a closed state bound to an inhibitory nanobody. Now, the present article by Basak et al. describes the structure of 5-HT_{3A} receptor in the resting apo-state determined by single particle cryo-EM. This furthers our understanding of the pentameric ligand-gated ion channels in general and 5-HT_{3A} receptor in particular. The authors have compared well the two structures and highlighted the differences between the two in the accompanying figures. An advancement from the earlier structure is the electron density for the complete M2-M3 linker which was only partially resolved in the 5-HT_{3AR} crystal structure. Although the resolution at 4.3 Å is not sufficient to see the side chains, it is enough for tracing the backbone. Another significant point is that the structure was determined by full receptor protein including the intracellular domain whereas to obtain the crystal structure a region of the intracellular domain was trypsinized. The absence of electron density of this region in the cryo-EM structure now confirms the unstructured nature of the intracellular domain region between MX and MA helices. Overall, the article is written well and with the use of clear language.

Overall, this apo cryo-EM structure resembles the nanobody-bound X-ray structure, establishing that both correspond to a resting-state like conformation, a conclusion that was not evident from the X-ray structure alone. In addition, the comparison of the two structures show significant differences, especially a quaternary compaction of the ECD and some motion of the M3 and M4 helices. This gives insights into the intrinsic dynamics of the resting state and further documents that antagonist-bound resting state might deviate from the apo state, a feature that might have interesting implication in the molecular mechanism of orthosteric inhibition.

Minor point:

Authors should specify in the result section that the structure was that of a detergent-solubilized protein

P6, line 5/6: please explain in more detailed " the axis through the intrasubunit cavity."

Reviewer #2 (Remarks to the Author):

This manuscript describes the cryo-EM structure of the mouse 5HT_{3A} receptor. It represents the first structure determination in this family from full-length receptor protein. This work is important even if largely confirmatory, because the previous X-ray structure was obtained after trypsin digestion and in the presence of a nanobody that stabilized the receptor in an ambiguous conformational state. This new apo structure thus provides a robust template for probing structural principles related to the resting state. The journal seems appropriate for this work. There are many significant concerns relating to over interpretation and lack of clarity in analysis, however these issues are correctable.

Major:

Abstract: this is not a full-length structure. It is the structure from full-length protein. Precise language will not diminish impact.

Final sentence in abstract highlights importance of inter-subunit interactions- but these are hardly discussed in the manuscript. Focus on key points here.

The introduction (second paragraph) suggests that this manuscript will answer the question of how the modular domains (ECD, TMD and ICD) influence the function of each other, and how the ICD regulates channel gating, in a manner that heretofore has been impossible. This manuscript is confirmatory in a very important way and usefully descriptive, but it does not elucidate a gating mechanism or new allosteric connectivities that were not understood until now. Please just tone down the language a bit.

In the introduction, please provide a little discussion of the fact that these channels activate and desensitize- discuss the gating cycle, and why you think the apo resting state is distinct from an antagonist bound state.

Fig. 1, FSC analysis of model vs. half maps used and not used in refinement should be shown to test for overfitting. See PMIDs 24675956, 29019981 and 28002411 as examples. Map vs. model FSC should be assessed at FSC = 0.5, not 0.143.

Fig. 1, dose response curve must have a data point near the middle of the response in order to trust the hill coefficient, and 5 concentration points is insufficient; 7 is a bare minimum.

In Table 1, please include Molprobity Clashscore with percentile and also percentile for overall Molprobity score. Please also include average B factors of protein, ions, waters and sugars and phospholipid for the refined model. Spell check Table 1. Ramachandran stats are significantly worse than in the x-ray structure- they should be the same or better, as that was the starting model, and the EM structure is at lower resolution. Suggests restraints were too loose in Refmac.

p. 6 comparison of buried surface areas: discussion of this measurement needs to take into account different modeled construct lengths or it is not meaningful. Also, you describe an expansion in the new Apo structure, which I would naively have assumed would correlate with less surface area buried (fewer close contacts at interfaces). Please elaborate on your interpretation of the surface analysis.

Fig. 3b: M2M3 loop appears to be in a completely novel conformation. This substructure is critical in gating- please discuss. Does it look more like in Unwin's Torpedo structures than all other structures in the superfamily? This is a major finding. Or, maybe I am wrong about interpreting the M2M3 loop conformation in Fig. 3 and SF 4.

Fig. 4: Are the densities for K108 and R416 clearly resolved? I understand that you will not expect to see density for the E250 in an EM map, but what about the other residues that you show creating constriction points? These could be shown in a small supplementary figure. These are major contrasting regions with that observed in the x-ray structure, based on Fig 4b. Also, are the water/ion density peaks visible in non-symmetry-averaged reconstructions? If so, that is a strong argument that they are real. Also as mentioned comparing B factors for these ions/waters would solidify the argument that they are real and correctly modeled.

In the conclusion it is an overstatement to say that constrictions can now be unambiguously assigned - this is still not a high resolution structure, there is no density for the glutamates at the selectivity

filter, and I did not see a density map figure for the other constriction points.

Some parts of the structure indeed look expanded, but some parts superimpose perfectly with the X-ray structure- is it really expanded, or is it more simply twisted?

Conclusion: generally I think of inhibited (competitive-antagonist-bound) and resting states as conformationally identical, at least for channels. Your ending suggests that you view these as distinct states and distinct conformations. Please elaborate a bit.

Minor:

Introduction:

Most serotonin receptors are GPCRs- the intro is currently written implying that all serotonin receptors are ion channels.

Last sentence in first paragraph: it does not obviously follow from all the clinical descriptions that a structure will help with all these problems. The last sentence needs more of a lead in or somehow be made less abrupt following the preceding sentences.

First sentence of second intro paragraph: please write more clearly that 5-HT₃ receptors can assemble as homopentamers of A subunits or as heteropentamers that include an A subunit as well as some combination of B-E subunits. As it stands this first sentence is confusing.

Second intro paragraph, "intracellular cytoplasmic domain...". Intracellular and cytoplasmic are redundant terms.

Supp Fig 1: Was the second (final) 3D classification actually important/useful- did it improve final resolution compared to skipping the final 3D classification step?

Page 5, end of first paragraph, modeled amino acid boundaries are listed. Please also list how many more/less residues this is than the X-ray structure.

Fig. 1 could move to Supplementary Materials.

Fig. 1B, what are the Y-axis units and please show a gel of sample used for EM.

Was PNGase treatment important? It looks like no sugars were cleaved.

Densities of glycans section on p. 5: there are many structures available in this family (GluCl, AChBP, nicotinic receptor...) that have density for glycans. As written, the implication is that only this new structure and the GABA-A homopentamer have resolved sugars. Also: is there anything to be learned about mechanism here? If not, they should not be a huge point of emphasis in Fig 2.

Fig. 3 legend: For the "right" panels, please be explicit about what is being superimposed for comparison. 3b: what do the different colored dashed lines indicate (what do the different colors mean, because these colors are not the same colors as the models)? Please also indicate somewhere which atoms were used as reference points. 3b: please label all helices.

Fig. 3: For the "left" panels, do I understand correctly that superpositions were made for panel A of the TMDs of the next subunit in the pentamer to compare the ECDs of the preceding subunit? If so, why not superimpose the TMD bundles of the same subunit you are displaying? The logic of what was

superimposed and why would be helpful to explain a bit more.

p. 6, top paragraph: intersubunit cavity. The figure you refer to seems to be an intrasubunit cavity- please confirm, which do you mean? And is there really a cavity here, or is it filled with side chains?

p. 6 end of first paragraph, please remove the word "interestingly" as it weakens the statement.

p. 6 lower paragraph: "vestibule in the TMD". The vestibule is generally thought to be the large extracellular solvent-filled volume; to refer to a transmembrane vestibule is confusing.

p. 8 please clarify what is meant by "mid-M2."

Fig. 4: what is the bases for defining the radius of a hydrated Na⁺ ion at 3.0Å? As Hille says, hydration is a fuzzy concept. A reference here would be helpful.

Fig. 4: Panel A says L108; Panel B says K108. Looks like Panel A needs to be corrected.

Fig. 4a: please show L406.

Supp Fig. 3: please change some of the colors so you are not using 3 shades of blue, and can you please label the traces themselves? Also, the colors in the legend are inconsistent with the colors of the lines.

Supp Fig. 4b and text at end of p. 8 / start of p. 9: is this stacking interaction between M4 and Cys-loop observed in the 5HT3 crystal structure? If not, this is new and very interesting, could be discussed more.

Fig. 5a: background is too faded making it difficult orient viewer in the region in the structure.

Fig. 5a: W156 is flipped relative to all other structures at agonist binding sites? Is this rotamer change really justified by the density? If so, this interesting finding should be discussed.

Second to last sentence on p. 9: which structures specifically are you talking about here- when you say antagonist, are you talking about the nanobody-bound structure, or AChBP-bound structures, or something else? And same question for agonists.

Last sentence on p. 9: you have an entire figure and a paragraph of text making this comparison, so why then say here that you cannot make this comparison.

Fig 6a and Supp Fig 4: With a 4.3Å structure and no density for anionic side chains (because this is an EM map), it is probably not appropriate to draw dashed lines suggestive of H-bonds or salt bridges, unless you have independent validation of these interactions.

Methods: are the defocus ranges listed derived from the fitting, or from the input range settings for data collection? Is 130 k x mag the actual mag (based on calibration), or just that set on the microscope? Lastly, what were the 4 strep tags per subunit for?

Reviewer #3 (Remarks to the Author):

In the present manuscript, Basak and colleagues report the 3-dimensional structure of the 5-HT_{3A} receptor in its apo form as determined by cryo-electron microscopy. The X-ray crystal structure of the 5-HT_{3A} receptor was previously published in complex with a bound antibody fragment acting as a crystallization chaperone. The present study offers an important complement to the available data and brings us one step closer to obtaining individual snapshots of this highly dynamic channel protein, which exists in at least 3 conformational states: closed, open and desensitized. The present work convincingly reveals the channel in the closed/resting state.

I find that the data are very well presented and believe that the authors made a great effort in interpreting the structural data in the context of previously obtained structural, functional or mutagenesis data. The manuscript reads very well and I believe it is very accessible even to the non-expert audience. I have no major remarks and strongly support publication of the manuscript.

A few minor remarks;

- the term "neurogastrointestinal" is somewhat out of context. I would separate the psychiatric disorders from the gastrointestinal disorders.
- I would reverse the order of the clinical application of 5-HT₃ antagonists. The primary indication is anti-emesis after radio/chemotherapy. Only a few selected -setrons are approved other clinical indications.
- page 4. I prefer the term concentration-activation curve rather than dose-response.
- It would be useful to have a superposition of the ICD from the crystal structure in Figure 6.
- The EM table has a typo ("resolution") and it would be useful to have the Molprobity percentile score between brackets
- Most but not all references contain DOI's. This should be made uniform according to Nature Comm. standards.

Reviewers' comments:

We thank all three reviewers for their time to review our manuscript. We are pleased with the positive feedback and thankful for the insightful comments. We have incorporated all the suggested changes. We hope the reviewers will find the revision satisfactory and the manuscript acceptable for publication.

Reviewer #1 (Remarks to the Author):

In 2014 Hassaine et al. described the X-ray crystal structure of 5-HT_{3A} receptor in a closed state bound to an inhibitory nanobody. Now, the present article by Basak et al. describes the structure of 5-HT_{3A} receptor in the resting apo-state determined by single particle cryo-EM. This furthers our understanding of the pentameric ligand-gated ion channels in general and 5-HT_{3A} receptor in particular. The authors have compared well the two structures and highlighted the differences between the two in the accompanying figures. An advancement from the earlier structure is the electron density for the complete M2-M3 linker which was only partially resolved in the 5-HT_{3AR} crystal structure. Although the resolution at 4.3 Å is not sufficient to see the side chains, it is enough for tracing the backbone. Another significant point is that the structure was determined by full receptor protein including the intracellular domain whereas to obtain the crystal structure a region of the intracellular domain was trypsinized. The absence of electron density of this region in the cryo-EM structure now confirms the unstructured nature of the intracellular domain region between MX and MA helices. Overall, the article is written well and with the use of clear language.

Overall, this apo cryo-EM structure resembles the nonabody-bound X-ray structure, establishing that both correspond to a resting-state like conformation, a conclusion that was not evident from the X-ray structure alone. In addition, the comparison of the two structures show significant differences, especially a quaternary compaction of the ECD and some motion of the M3 and M4 helices. This gives insights into the intrinsic dynamics of the resting state and further documents that antagonist-bound resting state might deviate from the apo state, a feature that might have interesting implication in the molecular mechanism of orthosteric inhibition.

Minor point:

Authors should specify in the result section that the structure was that of a detergent-solubilized protein

We now mention it in the results section. “Three dimensional reconstructions of the detergent-solubilized, full-length 5-HT_{3AR} channel led to a density map with a nominal resolution of 4.3 Å with imposed C₅ symmetry”

P6, line 5/6: please explain in more detailed “ the axis through the intrasubunit cavity.”

“The TMD helices, particularly M3 and M4, show an outward displacement with respect to an axis through the intrasubunit helical bundle composed of M1-M4 helices (Fig. 2B). However, there is very little change in the position of the M2 helix. Changes in M3 and M4 conformations relative to M2 lead to a widening of the intrasubunit cavity lined by M1-M4 helices (indicated by dotted lines, Fig. 2B).”

Reviewer #2 (Remarks to the Author):

This manuscript describes the cryo-EM structure of the mouse 5HT_{3A} receptor. It represents the first structure determination in this family from full-length receptor protein. This work is important even if largely confirmatory, because the previous X-ray structure was obtained after trypsin digestion and in the presence of a nanobody that stabilized the receptor in an ambiguous conformational state. This new apo structure thus provides a robust template for probing structural principles related to the resting state. The journal seems appropriate for this work. There are many significant concerns relating to over interpretation and lack of clarity in analysis, however these issues are correctable.

Major:

Abstract: this is not a full-length structure. It is the structure from full-length protein. Precise language will not diminish impact.

Edited as suggested. “Here, we present the structure from the full-length 5-HT_{3A}R channel in the apo-state determined by single-particle cryo-electron microscopy at a nominal resolution of 4.3 Å.”

The change has also been made in the main text. We removed “full-length” from the title.

Final sentence in abstract highlights importance of inter-subunit interactions- but these are hardly discussed in the manuscript. Focus on key points here.

As suggested, we have removed this sentence from the abstract focusing on the key points.

The introduction (second paragraph) suggests that this manuscript will answer the question of how the modular domains (ECD, TMD and ICD) influence the function of each other, and

how the ICD regulates channel gating, in a manner that heretofore has been impossible. This manuscript is confirmatory in a very important way and usefully descriptive, but it does not elucidate a gating mechanism or new allosteric connectivities that were not understood until now. Please just tone down the language a bit.

We agree with the reviewer that this structure represents just one of the many conformational states and to describe the structural changes underlying gating we need structures of other conformational states. We have removed the lines stating gating mechanism and the mechanism by which ICD influences channel gating.

In the introduction, please provide a little discussion of the fact that these channels activate and desensitize- discuss the gating cycle, and why you think the apo resting state is distinct from an antagonist bound state.

Per suggestion the introduction includes the following descriptions

“At the functional level, the 5-HT_{3A}R gating cycle involves transitions between at least three distinct conformational states namely; the resting state, a non-conductive conformation with low agonist-affinity; the open state, a conductive conformation with higher agonist-affinity; and the desensitized state, a non-conductive conformation, with the highest agonist-affinity among the three states. In the absence of the agonist (serotonin), the channel resides predominantly in the resting or closed conformation, while in the presence of the agonist, the channel transiently opens and eventually transitions to the desensitized conformation. Several therapeutically interesting compounds act as orthosteric or allosteric ligands and modulate 5-HT_{3A}R channel activity by shifting the equilibrium between these preformed functional states.”

Regarding apo state being distinct from antagonist bound state

“Although the inhibited state (at least in the case of competitive antagonism) could be expected to resemble the resting (closed) conformation, experimental evidences from voltage-clamp fluorometry, X-ray crystallography, and theoretical predictions in pLGIC homologues suggest that the inhibitors elicit conformational changes of their own even though the channel is electrically silent²³⁻²⁷. Interestingly, the antagonist-induced structural changes at some positions are similar while others distinct to those evoked by the agonist suggesting that the antagonist stabilizes the receptor in a different non-conductive conformation^{23,25}. Thus, to determine the basic tenets of the resting conformation, we sought to determine the structure of the full-length 5-HT_{3A}R channel in the apo-conformation by single-particle cryo-electron microscopy (Cryo-EM).”

Fig. 1, FSC analysis of model vs. half maps used and not used in refinement should be shown to test for overfitting. See PMIDs 24675956, 29019981 and 28002411 as examples. Map vs. model FSC should be assessed at FSC = 0.5, not 0.143.

Please see the refined model vs half maps and summed map information in supplemental Figure 2. We apologize for this omission.

Fig. 1, dose response curve must have a data point near the middle of the response in order to trust the hill coefficient, and 5 concentration points is insufficient; 7 is a bare minimum.

The dose response plot now includes additional concentration points (Supplemental Figure 1A).

In Table 1, please include Molprobity Clashscore with percentile and also percentile for overall Molprobity score. Please also include average B factors of protein, ions, waters and sugars and phospholipid for the refined model. Spell check Table 1. Ramachandran stats are significantly worse than in the x-ray structure- they should be the same or better, as that was the starting model, and the EM structure is at lower resolution. Suggests restraints were too loose in Refmac.

Thank you for the suggestion. We have carried out refinements with more stringent restraints. The Table 1 has updated refinement statistics. The table now includes the B-factors for proteins, glycans, phospholipid, ions, and water.

p. 6 comparison of buried surface areas: discussion of this measurement needs to take into account different modeled construct lengths or it is not meaningful. Also, you describe an expansion in the new Apo structure, which I would naively have assumed would correlate with less surface area buried (fewer close contacts at interfaces). Please elaborate on your interpretation of the surface analysis.

Now included in Methods “To compare apo-5HT_{3A}R and the 5-HT_{3A}R crystal structure, all ligands, ion, water molecules, and nanobodies (in the crystal structure) were removed from the PDB files. Additional residues in the apo-5HT_{3A}R structure were also removed before analysis so that surface area comparisons were made between identical construct lengths.”

“These structural differences are also reflected as an increase in total surface area (protein-only region) of the apo-5-HT_{3A}R pentamer (102, 850 Å²) in comparison to the crystal structure (85, 000 Å²). The increase in total surface area is accompanied by an increase in solvent accessible areas for apo-5-HT_{3A}R pentamer (73, 210 Å²) in comparison to the crystal structure (54, 530 Å²). Analysis of inter-subunit interfaces reveals that the buried areas between adjacent principal and complementary subunits are slightly reduced in apo-5-HT_{3A}R (2858 Å²) compared to the crystal structure (3045 Å²).”

Fig. 3b: M2M3 loop appears to be in a completely novel conformation. This substructure is critical in gating- please discuss. Does it look more like in Unwin’s Torpedo structures than

all other structures in the superfamily? This is a major finding. Or, maybe I am wrong about interpreting the M2M3 loop conformation in Fig. 3 and SF 4.

We have included a new panel in supplemental figure 7 which shows an alignment various pLGIC to highlight M2-M3 conformational differences. The text includes this:

“The M2-M3 linker is seen in multiple orientations in different pLGIC structures with this region adopting a distinct extended conformation in the apo-5HT_{3A}R (Supplemental Fig. 7B). The M2-M3 linker in the agonist-bound GlyR structure is retracted from the intersubunit interface in comparison to the antagonist-bound structure¹⁵. The M2-M3 linker and the pre-M1 region are implicated to be a part of the signal transduction machinery that communicates ligand-binding events in the ECD to the channel gate in the M2 bundle³⁸. Consequently, mutations in these regions are shown to impact channel function³⁹.”

Fig. 4: Are the densities for K108 and R416 clearly resolved? I understand that you will not expect to see density for the E250 in an EM map, but what about the other residues that you show creating constriction points? These could be shown in a small supplementary figure. These are major contrasting regions with that observed in the x-ray structure, based on Fig 4b. Also, are the water/ion density peaks visible in non-symmetry-averaged reconstructions? If so, that is a strong argument that they are real. Also as mentioned comparing B factors for these ions/waters would solidify the argument that they are real and correctly modeled.

As suggested, we have included the electron density map for constriction points along the permeation pathway in Supplemental Figure 5. Given the limitation of our resolution, the reconstructions were C5 symmetry averaged. We have now included the B-factors for water, ions, and lipids in Table 1. Based on these values, and on the distance between the modeled ions, water, and protein side chains, we suggest that these are potential ion binding sites.

In the conclusion it is an overstatement to say that constrictions can now be unambiguously assigned- this is still not a high resolution structure, there is no density for the glutamates at the selectivity filter, and I did not see a density map figure for the other constriction points.

Removed “unambiguously assigned”. Changed to “In addition, the locations of constriction that occlude ion permeation in the resting conformation can be inferred. A new supplemental figure is included to show the density maps at the constriction points Supplemental Figure 5.

Some parts of the structure indeed look expanded, but some parts superimpose perfectly with the X-ray structure- is it really expanded, or is it more simply twisted?

It appears to be a combination of twist and expansion. We have clarified these changes. “An alignment of the two structures reveals an expansion of all the three domains in apo-5-HT_{3A}R resulting from a radial outward twist of the ECD outer β -sheet and an iris-like movement of the TM helices and the ICD (Fig 2A). The ECD undergoes a clockwise twist with respect to the crystal

structure, which changes the relative positions of the interfacial loops at the ECD-ECD and ECD-TMD interfaces (Fig. 2B). Particularly, the movement in the outer β -strands is more pronounced than the inner β -strands. In the TMD, the M3 and M4 helices show an outward displacement with respect to an axis through the intrasubunit helical bundle composed of M1-M4 helices, while there is very little change in the position of the M2 helix (Fig. 2B). Changes in M3 and M4 conformations relative to M2 lead to a widening of the intrasubunit cavity lined by M1-M4 helices (indicated by dotted lines, Fig. 2B)."

Conclusion: generally I think of inhibited (competitive-antagonist-bound) and resting states as conformationally identical, at least for channels. Your ending suggests that you view these as distinct states and distinct conformations. Please elaborate a bit.

In a simple view a competitive antagonist occupies the same or overlapping binding site and prevents agonist from binding, thereby stabilizes the closed (resting) conformation. Alternatively, the antagonist, in addition to exerting steric hindrance to agonist binding, may also induce a structural change in the channel, perhaps opposite to that evoked by the agonist, hence stabilizes the receptor in a distinct non-conductive conformation. There are several lines of evidences that seem to suggest that the inhibitor bound state may differ from the resting conformation, both in structure and dynamics.

Included in the introduction.

"Although the inhibited state (at least in the case of competitive antagonism) could be expected to resemble the resting (closed) conformation, experimental evidences from voltage-clamp fluorometry, X-ray crystallography, and theoretical predictions in pLGIC homologues suggest that the inhibitors elicit conformational changes of their own even though the channel is electrically silent²³⁻²⁷. Interestingly, the antagonist-induced structural changes at some positions are similar while others distinct to those evoked by the agonist suggesting that the antagonist stabilizes the receptor in a different non-conductive conformation^{23,25}. Thus, to determine the basic tenets of the resting conformation, we sought to determine the structure of the full-length 5-HT_{3A}R channel in the apo-conformation by single-particle cryo-electron microscopy (Cryo-EM)."

Minor:

Introduction:

Most serotonin receptors are GPCRs- the intro is currently written implying that all serotonin receptors are ion channels.

Changed to "The ion channel class of serotonin receptors (5-HT_{3A}R) are cation selective and belong to the super family of pentameric ligand-gated ion channels (pLGIC)."

Last sentence in first paragraph: it does not obviously follow from all the clinical descriptions

that a structure will help with all these problems. The last sentence needs more of a lead in or somehow be made less abrupt following the preceding sentences.

We re-organized the two paragraphs for a better flow.

First sentence of second intro paragraph: please write more clearly that 5-HT₃ receptors can assemble as homopentamers of A subunits or as heteropentamers that include an A subunit as well as some combination of B-E subunits. As it stands this first sentence is confusing.

“The 5-HT_{3A}R pentameric complex is ~250 KDa in molecular weight and made up of five homologous subunit A (5-HT_{3A}) or a heterologous combination of subunit A with other subunits (B-E) arranged around a pseudo five-fold symmetric axis³.”

Second intro paragraph, “intracellular cytoplasmic domain...”. Intracellular and cytoplasmic are redundant terms.

Changed to “intracellular domain”.

Supp Fig 1: Was the second (final) 3D classification actually important/useful- did it improve final resolution compared to skipping the final 3D classification step?

Yes, although it did not improve the resolution, the quality of the final map was better.

Page 5, end of first paragraph, modeled amino acid boundaries are listed. Please also list how many more/less residues this is than the X-ray structure.

We added this information in Methods where we discuss model building “In comparison to the 5-HT_{3A}R crystal structure, for each monomer, 15 additional residues were built in our model and side-chains were built for 7 additional residues.”

Fig. 1 could move to Supplementary Materials.

Per suggestion, Figure 1 (now Supplemental Figures 1 and 2) are in the Supplementary.

Fig. 1B, what are the Y-axis units and please show a gel of sample used for EM.

Y-axis unit is UV Absorbance (text added to the figure). Gel picture included in Supplemental Figure 1.

Was PNGase treatment important? It looks like no sugars were cleaved.

Yes, it is a good point. We haven't looked into the sample without the PNGase treatment.

Densities of glycans section on p. 5: there are many structures available in this family (GluCl, AChBP, nicotinic receptor...) that have density for glycans. As written, the implication is that only this new structure and the GABA-A homopentamer have resolved sugars. Also: is there anything to be learned about mechanism here? If not, they should not be a huge point of emphasis in Fig 2.

Removed the arrows highlighting the glycans, included references for resolved glycans in other structures, and shortened the discussion here to reduce emphasis.

Fig. 3 legend: For the “right” panels, please be explicit about what is being superimposed for comparison. 3b: what do the different colored dashed lines indicate (what do the different colors mean, because these colors are not the same colors as the models)? Please also indicate somewhere which atoms were used as reference points. 3b: please label all helices. Fig. 3: For the “left” panels, do I understand correctly that superpositions were made for panel A of the TMDs of the next subunit in the pentamer to compare the ECDs of the preceding subunit? If so, why not superimpose the TMD bundles of the same subunit you are displaying? The logic of what was superimposed and why would be helpful to explain a bit more.

In the revision, this is Figure 2. The panels were rearranged for a better flow. The following changes are made as per suggestion. The top panels are aligned as “A view of the ECDs from the extracellular end when aligned with respect to the TMDs (left)” “The view of the TMD from the intracellular end when aligned with respect to the ECDs (right).” We changed the dotted line colors to match the colors of the structural models. Helices are labeled and the residues used as reference points are indicated (As circles in the figures and mentioned in the figure legends).

For the bottom panels showing i) ECD (left), the alignment was made with respect to the TMD of the adjacent subunit ii) TMD (right), the alignment was made with respect to the ECD of the adjacent subunit. This alignment strategy was chosen to highlight the relative changes both respect to the neighboring subunit as well as with respect to the other domain. This is included in the figure legend.

p. 6, top paragraph: intersubunit cavity. The figure you refer to seems to be an intrasubunit cavity- please confirm, which do you mean? And is there really a cavity here, or is it filled with side chains?

Thank you for pointing this. We changed this to intrasubunit cavity.

p. 6 end of first paragraph, please remove the work “interestingly” as it weakens the statement.

Changed.

p. 6 lower paragraph: “vestibule in the TMD”. The vestibule is generally thought to be the large extracellular solvent-filled volume; to refer to a transmembrane vestibule is confusing.

Changed to pathway instead.

p. 8 please clarify what is meant by “mid-M2.”

Changed to “In contrast, residues below the 13’ are accessible in the open but not in the closed state.”

Fig. 4: what is the bases for defining the radius of a hydrated Na⁺ ion at 3.0Å? As Hille says, hydration is a fuzzy concept. A reference here would be helpful.

We now say the value as approximate considering that this is not a fixed value. Reference included in the main text as well as in the figure legend.

Fig. 4: Panel A says L108; Panel B says K108. Looks like Panel A needs to be corrected. Fig. 4a: please show L406.

Changed. Please see Figure 3.

Supp Fig. 3: please change some of the colors so you are not using 3 shades of blue, and can you please label the traces themselves? Also, the colors in the legend are inconsistent with the colors of the lines.

Changed.

Supp Fig. 4b and text at end of p. 8 / start of p. 9: is this stacking interaction between M4 and Cys-loop observed in the 5HT3 crystal structure? If not, this is new and very interesting, could be discussed more.

The density for the tip of M4 was relatively weak in the crystal structure and Trp residue was built only in two of the subunits with the stacking interaction seen in one. We state this in the text now.

Fig. 5a: background is too faded making it difficult orient viewer in the region in the structure.

We changed the contrast and transparency to improve the quality of this figure.

Fig. 5a: W156 is flipped relative to all other structures at agonist binding sites? Is this rotamer change really justified by the density? If so, this interesting finding should be discussed.

This is a great point! The orientation of Trp156 in apo-5HT_{3A}R fits well in the density and is indeed flipped relative to its position in the 5-HT_{3A}R crystal structure. However, an alignment with AChBP, and nAChR (this position is a Tyr in anionic members) shows that the Trp156 orientation in apo-5HT_{3A}R matches with these structures instead (Supplemental Figure 9).

Second to last sentence on p. 9: which structures specifically are you talking about here-when you say antagonist, are you talking about the nanobody-bound structure, or AChBP-bound structures, or something else? And same question for agonists. We have clarified this “The Loop C conformation has been implicated to correlate both with the occupancy of the ligand at the neurotransmitter binding site and the functional state of the channel⁴⁴. Particularly in AChBP, in comparison to the apo-state, in the antagonist-bound state, Loop C adopts an “open” conformation, while in the agonist bound state it takes a “closed” conformation.”

Last sentence on p. 9: you have an entire figure and a paragraph of text making this comparison, so why then say here that you cannot make this comparison.

We have rephrased this sentence “A partly closed conformation of loop C in the crystal structure is likely from the nanobody bound in the vicinity.”

6a and Supp Fig 4: With a 4.3Å structure and no density for anionic side chains (because this is an EM map), it is probably not appropriate to draw dashed lines suggestive of H-bonds or salt bridges, unless you have independent validation of these interactions. We have removed the dashed lines.

Methods: are the defocus ranges listed derived from the fitting, or from the input range settings for data collection? Is 130 k x mag the actual mag (based on calibration), or just that set on the microscope? Lastly, what were the 4 strep tags per subunit for?

Defocus range: input range setting for data collection.

130kx mag: set on microscope.

The four strep tags were included in the construct for enabling additional purification strategies if needed.

Reviewer #3 (Remarks to the Author):

In the present manuscript, Basak and colleagues report the 3-dimensional structure of the 5-HT_{3A} receptor in its apo form as determined by cryo-electron microscopy. The X-ray crystal structure of the 5-HT_{3A} receptor was previously published in complex with a bound antibody fragment acting as a crystallization chaperone. The present study offers an important complement to the available data and brings us one step closer to obtaining individual snapshots of this highly dynamic channel protein, which exists in at least 3 conformational states: closed, open and desensitized. The present work convincingly reveals the channel in the closed/resting state.

I find that the data are very well presented and believe that the authors made a great effort in interpreting the structural data in the context of previously obtained structural, functional or mutagenesis data. The manuscript reads very well and I believe it is very accessible even to the non-expert audience. I have no major remarks and strongly support publication of the manuscript.

A few minor remarks;

- the term "neurogastrointestinal" is somewhat out of context. I would separate the psychiatric disorders from the gastrointestinal disorders.

Rephrased to "psychiatric and gastrointestinal disorders."

- I would reverse the order of the clinical application of 5-HT₃ antagonists. The primary indication is anti-emesis after radio/chemotherapy. Only a few selected -setrons are approved other clinical indications.

Rearranged as "Currently, serotonin receptor (5-HT_{3A}R) antagonists are in clinical use to alleviate nausea and vomiting caused by chemotherapy and radiotherapy, and for the management of post-infection diarrhoea and irritable bowel syndrome."

- page 4. I prefer the term concentration-activation curve rather than dose-response.

Change made as suggested.

- It would be useful to have a superposition of the ICD from the crystal structure in Figure 6.

An ICD superposition is now included in Figure 5A.

- The EM table has a typo ("resolution") and it would be useful to have the Molprobity percentile score between brackets

We apologize for the typo. Changes made per suggestion.

- Most but not all references contain DOI's. This should be made uniform according to Nature Comm. standards.

Thank you for pointing this out. For consistency, we have removed the DOI information for the references.

Reviewers' Comments:

Reviewer #2:

Remarks to the Author:

The authors thoroughly and satisfactorily addressed my concerns- the quick turnaround and thoughtful revision is much appreciated. Excellent work. I have a couple minor comments below, but do not need to see the MS again.

Minor:

Please clarify in Supp Fig 7 legend how the superposition was done- was one subunit used as the basis for superposition, or a region of one subunit, or the whole pentamer?

Reviewer 3 suggested changing the "dose-response" terminology- I disagree; "dose-response" is the commonly accepted terminology for this kind of experiment in this field. I do not feel incredibly strongly, however- please do what you prefer. Every reader will know what you mean.

Changes made per Reviewer 2's suggestions:

The authors thoroughly and satisfactorily addressed my concerns- the quick turnaround and thoughtful revision is much appreciated. Excellent work. I have a couple minor comments below, but do not need to see the MS again.

Minor:

- Please clarify in Supp Fig 7 legend how the superposition was done- was one subunit used as the basis for superposition, or a region of one subunit, or the whole pentamer?

We included this information. "A comparison of the position of M2-M3 linker as seen in various pLGIC structures (PDB-IDs: Styr-GlyR: 3JAD¹⁵; Apo-GluCl: 4TNV³⁰; Nic-nAChR: 5KXI¹³; Apo-nAChR: 2BG9¹²; and His-GABA- β 3: 4COF¹⁷). The principal subunit from these structures are superimposed on to apo-5-HT_{3A}R."

- Reviewer 3 suggested changing the "dose-response" terminology- I disagree; "dose-response" is the commonly accepted terminology for this kind of experiment in this field. I do not feel incredibly strongly, however- please do what you prefer. Every reader will know what you mean.

We changed it back to "dose-response".